# Assessing Head Check Crack Growth by Eddy-Current Testing

Stefan Marschnig [1,*] , Markus Loidolt [1] , Dieter Knabl [1] , Alwine Steinecker [1] and Reinhard Popp [2]

1   Institute of Railway Engineering and Transport Economy, Graz University of Technology, 8010 Graz, Austria
2   ÖBB-Infrastruktur AG, 1020 Vienna, Austria
*   Correspondence: stefan.marschnig@tugraz.at; Tel.: +43-664-60-873 (ext. 6717)

**Abstract:** Managing head checks is a crucial task for an infrastructure manager as in case of deep cracks, rails can break and, thus, accidents might be the consequence. Many infrastructure managers use vehicle-mounted eddy-current testing for detecting cracks. This is sufficient for guaranteeing safe railway operation in applying a reactive maintenance regime removing cracks of a pre-defined depth. Moving this towards a predictive maintenance regime is only possible through assessing the crack growth. Establishing a stable deterioration function needs a sound data basis including a proper re-positioning of the test results of consecutive testing campaigns. This paper presents the results achieved from analysing the eddy-current testing campaigns of 10 years on a main line of the Austrian railway network and calculating a crack growth function as regression to field data. While it is possible to derive stable functions, the testing frequency needs to be shortened in order to move further to predictive maintenance.

**Keywords:** railway infrastructure; track; rails; forecasting; head checking; RCF; rail grinding; maintenance

## 1. Introduction

Rolling contact fatigue (RCF) is an ever-present problem on all types of rail systems. It is a dominant cause of maintenance and renewal for mainline railways, and it is a significant economic and safety challenge for commuter and underground lines too. It is being studied intensively around the world [1–3].

Both on the wheel and on the rail, there are different types of defects caused by RCF [1]. Some examples are:

- Head checks;
- Wheel tread cracking;
- Shelling of rail or wheel;
- Squats.

Rolling contact fatigue is a form of surface-initiated fatigue caused by the repeated contact stresses between the rolling elements, such as wheels and rails, over time. The repeated loading and unloading of the contact zone can cause the material to crack and eventually fail, leading to problems such as rail or wheel breakage, which is a major safety concern in the rail industry [2].

Head checks are typically caused by a combination of factors such as high axle loads, heavy traffic and changes in the temperature and humidity. Head checks are typically characterised by their small size and shallow depth and can be difficult to assess visually. However, if left untreated, they can continue to grow and eventually lead to more serious defects such as rail fractures, which can pose a safety risk to trains and passengers [4].

To prevent rolling contact fatigue, rail and wheel materials must be designed to be fatigue resistant and maintenance procedures must be in place to detect and address any potential problems. This can include regular inspections of the track and rolling stock, as well as the use of materials known to be resistant to rolling contact fatigue [5,6].

There are several methods for detecting head checks in the rail, from visual inspection to eddy-current testing to ultrasonic testing. In the case of visual inspection, the identification of the rail damage is carried out by the inspector, usually during other tests [7].

Ultrasonic testing allows the detection of rail defects inside the rail without leaving any damage on the rail. The method is based on an impulse echo technique in which an ultrasonic signal penetrating the rail from the top is reflected at the base of the rail and sent back to the top of the rail. The time taken for the sound wave to travel is measured and compared with the calculated target time. If the sound wave is reflected at a crack, it will reach the top of the rail much earlier. However, this method can only determine the depth and not the length of the crack [7–9] and is limited to failures very close to the rail surface or at the gauge line.

Eddy-current inspection is primarily used on railway tracks to detect and evaluate head checks. This method is based on the electromagnetic interaction between the magnetic field of a probe and the currents induced in the specimen, with two to eight probes positioned between the running edge and the running surface of the rail. An alternating current flowing through a primary coil produces an alternating magnetic field. This induces eddy currents in the specimen, which generate their own secondary magnetic field. If a surface inhomogeneity occurs, the secondary magnetic field changes and superimposes on the primary field. Inhomogeneities can indicate cracks in the rail surface or changes in the material properties. The depth of the rail defects can only be calculated indirectly by assuming that the crack grows in depth at an angle $\alpha$ between 15° and 30° via the length of the crack and the crack angle. Due to the limited information provided by the ultrasonic method, which can only determine the depth of the crack, it is used in combination with the eddy-current method. This is particularly useful for detecting failures related to rolling contact fatigue [7–14].

In this paper, we take the eddy-current testing data of a main line of ÖBB, the Austrian infrastructure manager. ÖBB runs a corrective maintenance regime with RCF failure: when the crack depth reaches 1 mm, the affected section is foreseen to be maintained by grinding or milling within the next maintenance period. We note that the alert limit of 1 mm is very low so that this maintenance regime can still be considered to be preventive. We investigated if the eddy-current testing data are feasible to set up a stable deterioration function so that forecasting the point in time of reaching the alert limit is possible. This is the precondition for establishing a predictive maintenance regime.

## 2. Methodology

Data from an eddy-current testing vehicle running on the network of ÖBB-Infrastruktur AG was used for the evaluation. These evaluations are currently used for the preventive maintenance regime and are therefore considered trustworthy. Testing campaigns take place once a year for the evaluated track sections. Track sections with significant data gaps within the first campaign are measured a second time in the same year. In addition to the testing results, metadata on the track properties and maintenance measures carried out (grinding, milling) are available.

### 2.1. Data Preparation

To achieve meaningful results, proper data preparation was essential for time series analyses. For this reason, the first part of the methodology deals with data preparation. An initial step in data preparation was repositioning of the data. As can be seen in the upper part of Figure 1, the signal characteristics of the included testing campaigns were similar, but shifted in longitudinal direction to each other.

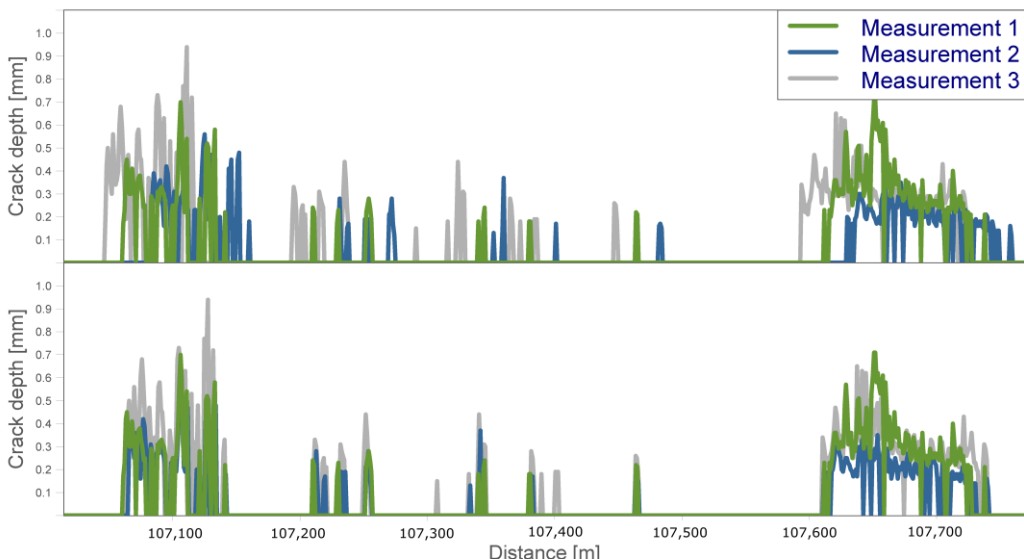

**Figure 1.** Crack measurements before and after repositioning.

This offset was caused by the inaccuracies in the positioning process implemented in the testing vehicle. When observing the signal, it became clear how the testing campaigns had to be shifted in order to achieve synchronicity. In principle, fine positioning of the data by means of correlation is conceivable; however, for this analysis, the shifts were made visually. It should be noted that it is not sufficient to shift testing campaigns as a whole, and that this procedure must be applied separately for each curve. The reason for this is an inaccuracy of the distance between two data points. This distance is 1 metre, but can deviate slightly, which is noticeable with signals that extend over kilometres, as in this evaluation.

The next step of the data preparation was linking the testing data with the track properties. Linkage is done on the basis of kilometre coding, as this is available for both types of data. A meaningful evaluation requires a sensible classification of the curves. Here, we included two parameters, radius and rail steel quality, as it could be assumed that, excluding the vehicle properties, these have the biggest influence on the formation of head checks considering that gross tonnage is constant for the analysed line section. Other track parameters such as sleeper type, ballast condition and subsoil quality vary over the line stretch but should not have high influence on the rail surface topic in a first approach. Table 1 gives an overview on the clustering of the curves. We only analysed the radii classes 2, 3 and 4, as we did not expect head checks in the sharp curves facing dominant wear and in tangent tracks. The borders of these radii classes had been set earlier for the ÖBB tracks when covering average maintenance frequencies.

**Table 1.** Existing parameter combination within the analysed section.

| Clustering of Curves | Abbreviation |
| --- | --- |
| Radii Class 1: R < 400 m | RC1 |
| Radii Class 2: 400 m < R < 600 m | RC2 |
| Radii Class 3: 600 m < R < 1000 m | RC3 |
| Radii Class 4: 1000 m < R < 3000 m | RC4 |
| Radii Class 5: R > 3000 m | RC5 |

In terms of steel grades, R260 is the standard grade used in Austria and, thus, the most frequent one in use. However, in curves, higher steel grades are used very frequently. R350HT and also R400HT are standard components in curves with excessive wear (radii below 400 m), but also in wider curves up to 3000 m against head checking, reaching some 30–40% of all the curves analysed for this paper. The calibration of the testing was done

with R260 steel grade. Therefore, the testing results of the higher steel grades (R350HT and R400HT) might not be comparable with their absolute values. We noticed that the number of curves not affected by head checks was higher for the higher steel grades, but the crack growth must be evaluated in detail in future research.

The final step of the data preparation was to define a quality index which could be used for the time series analysis. To make the data easier to interpret, the noise of the raw data was reduced. This was done by calculating the sliding mean of the signal, based on a window width of 5 m and a reduction interval of 1 m. If this calculation was applied to all testing campaigns of a curve, the signal characteristics result in Figure 2.

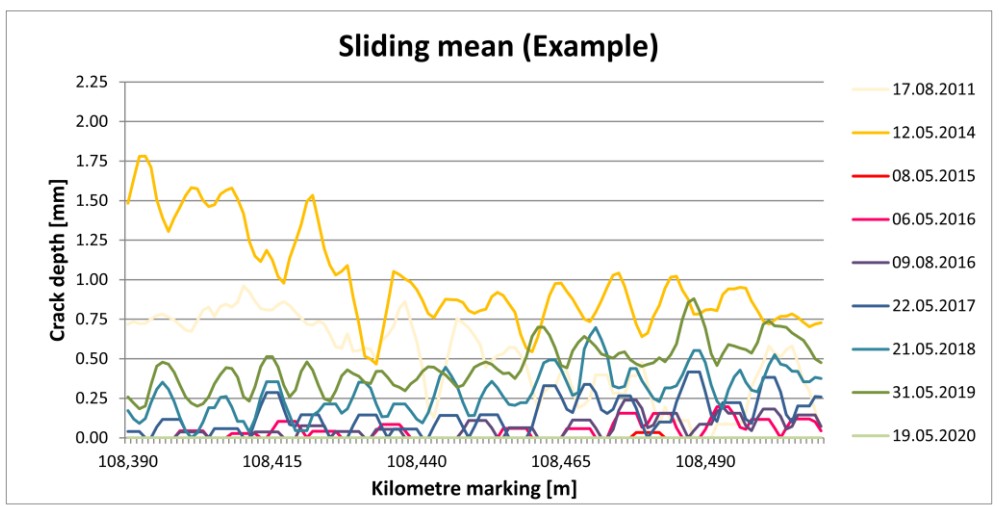

**Figure 2.** Sliding mean for different testing campaigns.

There are nine testing campaigns included in the figure, identifiable by their different colour. Seven of the campaigns show a similar characteristic with slightly deeper cracks at the end of the curve rather than at the beginning of the curve. Two campaigns (2011 and 2014) show a completely different characteristic with deeper cracks at the beginning of the curve. Simultaneously, the crack depth was generally higher for these testing campaigns. A further analysis of how crack growth differs within curves could lead to a better understanding of crack growth behaviour, but is not the subject of this paper. For the purpose pursued, which is the time series analysis, a further step of simplification was necessary, as one quality index per testing series was needed. Therefore, we then calculated the mean value of the signal over the entire curve. This led to an index (crack depth) for each testing campaign, which is shown in Figure 3 for the same curve as before.

**Figure 3.** Average crack depth per testing campaign.

Since the data in Figures 2 and 3 are the same, it becomes clear why two of the testing campaigns have a different characteristic than the others. The reason is the maintenance carried out between the testing campaigns in 2014 and 2015. This is also in line with ÖBB's maintenance regime, as cracks with a depth of more than 1 mm were detected during the campaign in 2014. Despite fully removing the cracks in the maintenance operation, they started to grow again and were first visible in the testing campaign in 2015. Thereby, crack growth does not occur linearly, but apparently exponentially over time. The example shows that eddy-current testing can yield stable time series if data are processed appropriately.

### 2.2. Time Series Analysis

After data preparation, the crack depth development as shown in Figure 3 was available for every analysed curve within the analysed section. For further processing, the curves had to be selected and sorted. As already mentioned, we separated the curves with different steel grades. According to the testing method, we analysed the rails with standard steel grade R260. This reduced the number of analysed curves from 118 to 66: 15 in RC2, 25 in RC3, and 26 in RC4. One curve in RC3 and two curves in RC4 were not affected by head checks. We defined three valid testing results in consecutive testing campaigns as quality criterion for a stable deterioration period. This removed another 14 curves, so that 52 curves finally defined the evaluation sample: 10 in RC2, 24 in RC3 and 18 in RC4.

Before a deterioration model could finally be applied, deterioration branches had to be defined. Besides the condition of three data points, we only used branches where a testing campaign could be clearly defined as the start of the deterioration branch. This was the case when a campaign with relevant crack depth was followed by a campaign with crack depths close to zero, meaning that it could be assumed that the maintenance or rail replacement had taken place between the testing campaigns. Data about executed maintenance action also provided information about the start of the deterioration branches. However, as this data was not always recorded completely, the final decision was made on basis of the testing itself. Since testing campaigns cover ten years, in some cases more than one deterioration branch could be assigned to a curve. Thus, 52 curves formed 73 deterioration branches.

From observing the data, it was obvious that the growth rates of head checks behaved in an exponential way; therefore, we applied an exponential model on the deterioration branches. Exponential growth is a mathematical model in which the initial value changes by a percentage in the same time steps. The left side of Figure 4 shows the theoretical model.

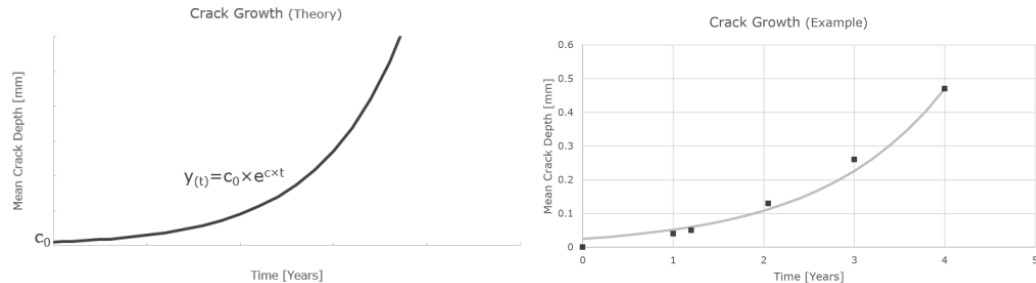

**Figure 4.** Exponential crack growth. Left: Theoretical model. Right: Model applied on data.

The parameters of the model were initial quality at the beginning of the deterioration process ($c_0$), growth rate (c) and time passed since the start of deterioration (t). By applying this model, it was essential to consider that the testing campaigns were carried out at different time intervals. However, as the exact point of time of the testing campaign was known, the time between the start of the deterioration rate and the respective testing campaign could be easily determined. Since the load collective did not change within the examined section, we have not included it in the evaluation. If further sections with other train collectives are considered, further considerations must be made here. The implementation of the model adjustment for the prepared deterioration branches was done with the software "Graph Pad Prism". After implementation, the descriptive parameters

of the exponential function were consequently available for every curve with the relevant parameter combination.

## 3. Results

In the first step, we calculated the crack growth regression function for all the selected deterioration periods. The left part of Figure 5 depicts the wide range of the values. Starting coefficients $c_0$ higher than 0.1 are not realistic as the cracks were totally removed within the grinding operation at ÖBB. Furthermore, we saw that the deterioration periods with missing data points in between give a high uncertainty; we cannot ensure that these periods have been maintenance free.

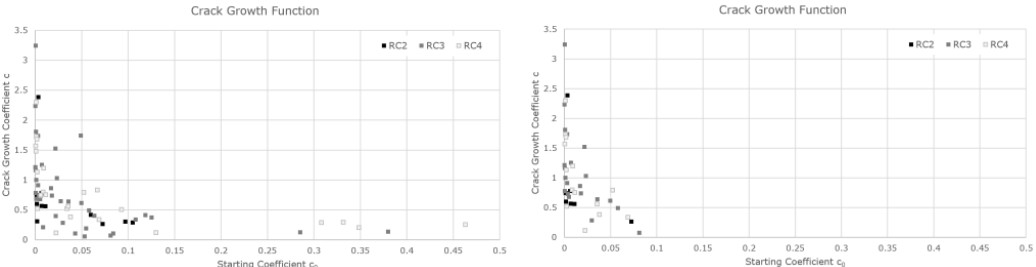

**Figure 5.** Exponential crack growth. Left: Theoretical model. Right: Model applied on data.

Thus, we continued the evaluation with only through-going testing results as additional criterion. This process decreased the number of analysed deterioration periods to 42. Consequently, the high starting coefficients disappeared after setting this filter (Figure 5, right).

Table 2 provides the median values of the starting coefficients and the crack growth coefficients for the three radii classes and the number of observations in the three groups.

**Table 2.** Median values of $c_0$ and c for the three radii classes.

|  | **RC1** | **RC2** | **RC3** |
|---|---|---|---|
| Number of Observations | 7 | 21 | 14 |
| Median starting coefficient $c_0$ | $6.255 \cdot 10^{-3}$ | $4.981 \cdot 10^{-3}$ | $8.782 \cdot 10^{-3}$ |
| Median crack growth coefficient c | $6.014 \cdot 10^{-1}$ | $9.106 \cdot 10^{-1}$ | $7.935 \cdot 10^{-1}$ |
| Average grinding interval 1.0 mm [years] | 8.4 | 5.8 | 6.0 |

Using an exponential function of the form $y(t) = c_0 \times ec \times t$, we needed to consider that the starting coefficient $c_0$ and the crack growth coefficient c needed to be analyzed jointly. Only if the starting coefficients are equal, the growth coefficients might be looked at separately. As the number of observations was low in our case, the starting coefficients differed significantly for the three radii classes (Table 2).

Nevertheless, we wanted to compare the crack growth in the different radii classes. Thus, we set a crack growth limit, in the first approach 1 mm (according to ÖBB's alert limit), and calculated the time span to reach the limit in the different radii classes. This result was equal to the average maintenance interval with the alert limit 1 mm. We see that the crack growth was very similar in radii classes 3 and 4 while radii class 2 showed a one third longer grinding interval. Due to the mentioned effects occurring in exponential deterioration functions, we also calculated the grinding interval for every single deterioration period and averaged those values for every radii class. This led to slightly different results in the mean values (not changing the ranking), but provided additionally the range of the observations (Figure 6, left).

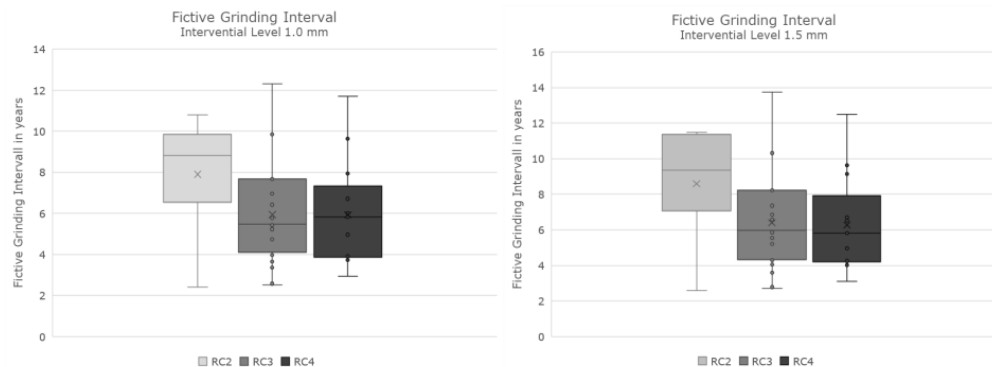

**Figure 6.** Fictive grinding interval for an alert limit of 1 mm (left) and 1.5 mm (right).

The literature often provides a certain crack growth rate per loading, mostly gross tonnage. Looking at the characteristic of the crack growth (exponential growth), we could not derive a constant growth rate from our results directly. In order to still compare our results with the literature values, we additionally calculated the fictive grinding interval for an alert limit of 1.5 mm (Figure 6, right) and 3 mm, and calculated the accumulated gross tonnage within these time periods. Table 3 gives the respective values.

**Table 3.** Average crack growth rates depending on the alert limit.

|  | RC1 | RC2 | RC3 |
|---|---|---|---|
| Number of Observations | 7 | 21 | 14 |
| **Alert limit 1.0 mm** | | | |
| Average grinding interval [years] * | 8.8 | 5.5 | 5.9 |
| Accumulated loading [$10^6$ gross-tonnes] | 1.61 | 1.00 | 1.08 |
| Average crack growth rate [mm/$10^6$ gross-tonnes] | 0.62 | 1.00 | 0.93 |
| **Alert limit 1.5 mm** | | | |
| Average grinding interval [years] | 9.3 | 6.0 | 6.2 |
| Accumulated loading [$10^6$ gross-tonnes] | 1.71 | 1.09 | 1.13 |
| Average crack growth rate [mm/$10^6$ gross-tonnes] | 0.88 | 1.38 | 1.33 |
| **Alert limit 3.0 mm** | | | |
| Average grinding interval [years] | 10.3 | 6.9 | 6.9 |
| Accumulated loading [$10^6$ gross-tonnes] | 1.88 | 1.27 | 1.26 |
| Average crack growth rate [mm/$10^6$ gross-tonnes] | 1.60 | 2.37 | 2.38 |

* Average of specific grinding intervals.

Depending on the alert limit, the average crack growth rate ranged between 0.62 and 2.38 mm/$10^6$ gross tonnes being in good accordance with the experience published by Deutsche Bahn [6]. We found that the crack growth rate for the curves with radii from 600 to 3000 m did not really differ, while in sharper curves it was significantly lower.

## 4. Conclusions

As we have shown, eddy-current testing can provide sufficient data to derive a function of crack growth. We managed it with the post-positioning of the data to calculate a crack growth function as regression to field data. The e-function shaped damage progress was within good understanding of the crack growth being low and flat in the beginning, and fast and steep having reached a certain depth. The results for different radii classes were also almost within common scientific sense, reaching higher crack growth rates for medium and wide curves while sharp curves were less affected. However, obviously there are many influencing factors apart from the curve radius as the crack growth rates strongly vary. It is within common sense that the vehicles play an important role, but in our case, the traffic composition is almost constant. Therefore, other factors occur.

We can state that the testing frequency, in our case one time per year, is not sufficient to move towards predictive maintenance in practical application. This is mainly due to the deterioration function itself; the forecasting becomes important when the crack starts to grow faster and, thus, moving into the steep part of the e-function. Therefore, more frequent testing results are needed to capture both the point in time of this change in the characteristic of the crack growth and the velocity of the crack growth in this part of the function. Moving towards a predictive maintenance regime requires a minimum of three testing campaigns per year.

## 5. Outlook

We also looked at higher rail steel grades within our research. We noticed that the testing results were valid for the standard steel grade being used in the calibration process of the testing equipment while delivering implausible results for higher steel grades.

Further results will focus on three main topics: (a) increasing the number of analysed curves in order to obtain more stability and, thus, reliability of the deterioration functions, (b) analysing the effect of loading starting with a similar gross tonnage composed by a very different vehicle-mix, and (c) investigating the effects occurring at higher rail steel grades.

**Author Contributions:** Conceptualization, S.M.; methodology, A.S., S.M. and M.L.; software, A.S.; validation, A.S., R.P. and S.M.; formal analysis, A.S.; investigation, A.S. and S.M.; resources, R.P. and S.M.; data curation, A.S.; writing—original draft preparation, S.M., M.L. and D.K.; writing—review and editing, S.M., M.L., D.K, R.P. and A.S.; visualization, S.M.; supervision, S.M., R.P. and M.L.; project administration, S.M.; funding acquisition, S.M. All authors have read and agreed to the published version of the manuscript.

**Funding:** Open Access Funding by the Graz University of Technology.

**Institutional Review Board Statement:** Not applicable.

**Informed Consent Statement:** Not applicable.

**Data Availability Statement:** Restrictions apply to this data. Data was obtained from ÖBB-Infrastruktur AG.

**Acknowledgments:** Data provided by ÖBB-Infrastruktur AG.

**Conflicts of Interest:** The authors declare no conflict of interest.

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
