# Peer review of "Assessing Head Check Crack Growth by Eddy-Current Testing"

_infrastructures, doi:10.3390/infrastructures8050089_

Round 1

Reviewer 1 Report

The submitted manuscript deals with the topic of research methods concerning the development of defects in railroad rails and is very interesting. The work is an important contribution to the field of maintenance and safety in rail transportation, it also directs the reader's attention towards predictive maintenance, which today is considered the optimal option. The structure of the work is correct, the literature is also appropriately selected in terms of quantity, quality and subject matter. A proportionately large number of references discuss the use of under sleeper pads, but this is just an observation, not a complaint. The language of the work is very good, although there are isolated glitches. Notwithstanding the above, I reccomend the changes, corrections and additions listed below before publication.

General remarks

1. The authors should stronger emphasize the aspect of novelty, innovation and distinguishing their work from previous achievements in the field.

2. The spelling of OeBB and ÖBB appears in various places in the manuscript, which should be standardized.

Detailed remarks

Line 41: to assess

Line 76: when (capital letter not necessary)

Fig. 4, 5: letters uncomfortably small to read, especially exponents

Line 217, 218: c[lower index]0

Line 242: Table 3

Table 3: if it’s written that accumulated loading equals 1.61, does it finally mean 161 million ton? Also, it is advised to avoid mio. abbreviation as understood properly only by German speakers, and to replace it with 10[power]6, mil. or mln. (also Line 244)

Line 246: ‘4.Discussion’ to be removed

Line 283 – 292: part of template, to be removed

Line 299: please expand ‘et al.’ to full author list

Author Response

see attached document

Reviewer 2 Report

The subject is interesting and current. The work provides a useful contribution to the search for models to be used for preventive maintenance.

Some suggestions

It would be useful to add a synthetic analysis of the state of the art relating to the numerous references listed in the bibliography.

Clarify whether parameters have been observed (geometry, speed, loads, etc.), which can justify the differences in growth between the small radius curves and the others.

Author Response

see attached document
